# Intra-Arterial Administration of Stem Cells and Exosomes for Central Nervous System Disease

**DOI:** 10.3390/ijms26157405

**Published:** 2025-07-31

**Authors:** Taishi Honda, Masahito Kawabori, Miki Fujimura

**Affiliations:** Department of Neurosurgery, Hokkaido University Graduate School of Medicine, Kita 15, Nishi 7, Kita-ku, Sapporo 060-8638, Hokkaido, Japan

**Keywords:** stem cell, exosome, extracellular vesicle, intraarterial, central nervous system disease

## Abstract

Central nervous system (CNS) disorders present significant therapeutic challenges due to the limited regenerative capacity of neural tissues, resulting in long-term disability for many patients. Consequently, the development of novel therapeutic strategies is urgently warranted. Stem cell therapies show considerable potential for mitigating brain damage and restoring neural connectivity, owing to their multifaceted properties, including anti-apoptotic, anti-inflammatory, neurogenic, and vasculogenic effects. Recent research has also identified exosomes—small vesicles enclosed by a lipid bilayer, secreted by stem cells—as a key mechanism underlying the therapeutic effects of stem cell therapies, and given their enhanced stability and superior blood–brain barrier permeability compared to the stem cells themselves, exosomes have emerged as a promising alternative treatment for CNS disorders. A key challenge in the application of both stem cell and exosome-based therapies for CNS diseases is the method of delivery. Currently, several routes are being investigated, including intracerebral, intrathecal, intravenous, intranasal, and intra-arterial administration. Intracerebral injection can deliver a substantial quantity of stem cells directly to the brain, but it carries the potential risk of inducing additional brain injury. Conversely, intravenous transplantation is minimally invasive but results in limited delivery of cells and exosomes to the brain, which may compromise the therapeutic efficacy. With advancements in catheter technology, intra-arterial administration of stem cells and exosomes has garnered increasing attention as a promising delivery strategy. This approach offers the advantage of delivering a significant number of stem cells and exosomes to the brain while minimizing the risk of additional brain damage. However, the investigation into the therapeutic potential of intra-arterial transplantation for CNS injury is still in its early stages. In this comprehensive review, we aim to summarize both basic and clinical research exploring the intra-arterial administration of stem cells and exosomes for the treatment of CNS diseases. Additionally, we will elucidate the underlying therapeutic mechanisms and provide insights into the future potential of this approach.

## 1. Introduction

Despite the advances of modern medicine, central nervous system (CNS) diseases, including stroke, trauma, psychiatric, and neurodegenerative diseases, are one of the intractable diseases that leave many patients disabled. There is an urgent need to develop a treatment method to treat these diseases. Stem cell therapies show considerable potential for mitigating brain damage and restoring neural connectivity, owing to their multifaceted properties, including anti-apoptotic, anti-inflammatory, neurogenic, and vasculogenic effects, and many clinical trials are currently ongoing to prove their safety and efficacy [1,2,3,4,5]. In recent years, considerable research efforts have also been directed towards elucidating the role of exosomes as a crucial therapeutic mechanism of stem cells. Exosomes, nano-sized vesicles ranging from 40 to 200 nm, are composed of a double lipid-layer membrane and harbor a plethora of molecules, including DNA, mRNA, microRNA (miRNA), and proteins [6]. Stem cell-derived exosomes have exhibited remarkable potential in mitigating central nervous system (CNS) diseases by virtue of their anti-apoptotic, anti-inflammatory, neurogenic, and angiogenic properties [7,8,9,10,11,12,13].

Regardless of their promising nature, a key challenge in the application of both stem cell and exosome-based therapies for CNS diseases is the method of delivery. Currently, several routes are being investigated, including intracerebral, intrathecal, intravenous, intranasal, and intra-arterial administration [11]. Intracerebral injection can deliver a substantial quantity of stem cells and exosomes directly to the brain, but it carries the potential risk of inducing additional brain injury. Intrathecal transplantation can also offer a large amount of stem cells and exosomes to the cerebroventricular space, while the efficacy of stem cell engraftment or exosome absorption is limited by the risk of hydrocephalus [14]. Intravenous transplantation is minimally invasive but results in limited delivery of cells and exosomes to the brain (1–10%), which may compromise the therapeutic efficacy [10,15,16,17]. The intranasal approach is also minimally invasive; however, its absorption rate is not as high as expected (<1%) [13]. Disappointing results from intravenous stem cell transplantation have driven the search for more effective therapeutic strategies to deliver stem cells to the damaged brain [18,19,20,21,22]. In these circumstances, with advancements in catheter technology, especially for treating ischemic stroke through thrombectomy, intra-arterial administration of stem cells and exosomes has garnered increasing attention as a promising delivery strategy. This approach offers the advantage of delivering a significant number of stem cells and exosomes to the brain while minimizing the risk of additional brain damage. However, the investigation into the therapeutic potential of intra-arterial transplantation for CNS injury is still in its early stages.

In this review, we summarize the intra-arterial administration of stem cells and their exosomes for CNS diseases, focusing on both basic and clinical research to clarify their efficacy and mechanism of action, as well as their limitations and future applications.

A literature search was conducted on PubMed (https://www.ncbi.nlm.nih.gov/pubmed, accessed on 25 February 2025) to identify research articles on intra-arterial stem cell or exosome administration for CNS diseases. The search utilized the keywords “intraarterial”, “stem cell”, “exosome or extracellular vesicle”, and “brain”. The articles included in the search were required to be written in English, relevant to CNS diseases, and specifically focused on stem cells and exosomes, while other articles were excluded. Additionally, we reviewed references cited within the selected papers from the preliminary search. The selection of articles and data collection were performed by one of the authors (T.H.). The collected data encompassed various disease models, cell and exosome sources, animal models, dosage, treatment duration, and labeling methods.

## 2. Overall Results of IA Transplantation of Stem Cell and Exosome for CNS Disease

A total of 87 articles were chosen that align with the objectives of this review; 71 articles were selected as preclinical studies that used stem cells via intra-arterial treatment (Appendix A) [23,24,25,26,27,28,29,30,31,32,33,34,35,36,37,38,39,40,41,42,43,44,45,46,47,48,49,50,51,52,53,54,55,56,57,58,59,60,61,62,63,64,65,66,67,68,69,70,71,72,73,74,75,76,77,78,79,80,81,82,83,84,85,86,87,88,89,90,91,92,93], 4 articles were selected as preclinical studies that used exosomes focusing on IA treatment (Appendix A) [94,95,96,97], and 12 articles were selected as clinical studies that used stem cells through IA treatment [98,99,100,101,102,103,104,105,106,107,108,109].

## 3. Preclinical Studies of Cell Therapy

Among the 71 studies reviewed, the majority (57 articles, 82%) focus on ischemic stroke. The remaining studies investigate traumatic brain injury, glioma, intracerebral hemorrhage, Alzheimer’s disease, Parkinson’s disease, and complication analysis associated with intra-arterial (IA) transplantation. This distribution is attributable to the fact that brain artery obstruction is the primary mechanism underlying ischemic stroke, making it logical to treat both the brain and blood vessels through the same therapeutic route. In this subsection, the authors summarize the preclinical data related to each of these diseases.

### 3.1. Ischemic Stroke

Ischemic stroke, caused by the occlusion of a brain blood vessel, leads to the deprivation of glucose and oxygen in its downstream branches, emerging as the leading cause of global disability. It is the third most common cause of death (10.7% of all deaths) and the fourth leading cause of Disability-Adjusted Life Years (DALYs) worldwide (5.6% of all DALYs) [110]. At present, standardized treatments, such as thrombectomy and recombinant tissue plasminogen activator (r-tPA) therapy, are implemented during the acute phase (<4.5 h) to re-establish blood flow in the occluded vessel and to salvage the penumbra. However, approximately 50% of the patients will present neurological deficits even after successful recanalization [111]. Consequently, intra-arterial transplantation of stem cells and exosomes is considered a promising approach to mitigate the severity of sequelae.

#### 3.1.1. Cell Sources

The source of stem cells is a critical factor in stem cell therapy (Figure 1, Appendix A). Of the 57 studies reviewed, 29 studies (51%) utilized human cells, while 28 studies (49%) employed animal allogeneic cells, with 1 study evaluating both human and animal allogeneic cells [51]. A wide range of human cell sources have been explored; 16 studies (52%) used bone marrow-derived mesenchymal stromal cells (BMSCs) or bone marrow-derived mononuclear cells (BMMNCs), 7 studies (22%) used umbilical cord blood-derived mesenchymal stem cells (MSCs), 3 studies (10%) used amnion-derived MSCs, 2 studies (6%) used adipose-derived MSCs and embryonic stem cells (ESC), and 1 study used hematopoietic stem cells, hair follicle stem cells, and induced pluripotent stem cells. In contrast, animal allogeneic cells demonstrated different characteristics. Eleven studies (40%) used BMSC or BMMNC, which are similar to human cells, while another eleven studies (40%) employed neural stem cells (NSCs) or neural crest-derived stem cells, which are derived from fetal brain tissue. The use of fetal brain-derived cells is particularly challenging, as they must be sourced from the fetal brain, which is not readily available for human applications. While most of the studies (42/58; 71%) utilized stem cells alone, some studies modified the cells through pretreatment with various substances, gene induction, and cell purification (Table 1). Interestingly, these modifications can be further categorized into three factors: enhanced neural differentiation [26,32,36,45,50,79,93], improved cell engraftment and migration to the damaged site [38,40,49,78], and the promotion of upregulating growth factor release or anti-inflammatory effects [23,24,85], all of which enhance the efficacy of cell therapy. Although most naïve cells show significant recovery, some studies compared modified cells with naïve cells. However, the modification of stem cells often results in higher labor costs and a lower cell collection yield, and the clinical value of such modifications remains uncertain.

Khabbal et al. compared the engraftment of xenogeneic human cells and allogeneic animal cells using isotope labeling techniques. They found that both cell types were detected in the ischemic brain shortly after transplantation, with radioactivity levels rapidly decreasing during the follow-up periods at 3 and 6 h. Interestingly, the radioactivity of xenogeneic human cells decreased much more rapidly than that of animal cells by 6 h after transplantation, which may be attributed to immune rejection [51]. This finding underscores the need for caution when extrapolating animal data to human studies. Salehi et al. compared neural crest stem cells (NCSCs) and BMSCs, finding that although both cell types exhibited satisfactory recovery, the expression of trophic factors differed significantly between the two. NCSCs expressed BDNF, nestin, Sox10, and doublecortin, while BMSCs expressed GDNF [34]. These findings highlight the importance of trophic factors in the recovery process, and further studies are needed to elucidate the specific trophic factors necessary for optimal recovery. Yamaguchi et al. compared young and old BMSC regarding the functional recovery and found that younger cells exhibited better neurogenesis and angiogenesis, highlighting the difference in cell donor age for recovery [39].

#### 3.1.2. Cell Doses

While the majority of studies provide animal weights in the Methods section, some articles do not. In such cases, we defined the rat weight as 250 g and the mouse weight as 25 g. The amount of cells administered intra-arterially varies widely across studies, ranging from 0.4 to 1333 × 10^5^ cells/kg (Figure 2, Appendix A). However, a clear trend emerges in which heavier animals receive fewer cells per body weight. Mice tend to receive approximately 200 × 10^5^ cells/kg, rats receive 50 × 10^5^ cells, and dogs receive 10 × 10^5^ cells/kg. This discrepancy may arise from the fact that most animals, regardless of weight, typically receive 1 × 10^6^ cells intra-arterially. There are several reports that focused on the therapeutic effect of the different cell doses. Fukuda et al. compared 0.4 and 36.4 × 10^5^ cells/kg of BMSC against the rat ischemic stroke model and found that both doses showed significant recovery with lower inflammation compared with the control group, and the high-dose group showed a higher mortality rate (39%) compared with the low-dose (27%) and control (33%) groups. They also observed a higher amount of microvessel clogging with the transplanted cells in the high-dose group [56]. When administering higher numbers of cells, there is always concern regarding microvessel obstruction, which can reduce blood flow and compromise recovery. Yavagal et al. compared cerebral blood flow following administration of various cell numbers (2–35 × 10^5^ cells/kg) and concluded that the maximum number of cells that does not compromise blood flow is 3.5 × 10^5^ cells/kg or less [60]. Cui et al. also reported that a lower dose (10 × 10^5^ cells/kg) was superior to higher doses (20 and 40 × 10^5^ cells/kg) in terms of minimizing blood flow reduction [58]. Magnetic resonance (MR) imaging studies also showed that the administration of a smaller number of cells (1 and 8 × 10^5^ cells/kg) resulted in less brain damage compared to a larger dose (26 × 10^5^ cells/kg), with larger doses leading to arterial clogging [44]. On the other hand, Greggio et al. compared 50 and 500 × 10^5^ cells/kg of umbilical cord blood mononuclear cells after perinatal ischemic brain injury and reported that a higher number of cells resulted in better functional recovery and smaller brain damage [59]. Yang et al. also reported that animals receiving 300 × 10^5^ cells/kg of bone marrow-derived mononuclear cells showed better recovery compared to those receiving 10 × 10^5^ cells/kg [70]. Wong et al. reported that interleukin-1a preconditioned MSCs (40 × 10^5^ cells/kg) enabled upregulation of blood flow monitored by Laser speckle imaging 1.5 h after cell transplantation, which they speculated was the role of vascular endothelial growth factor (VEGF) released from conditioned cells. In addition to the number of cells administered, the administration speed is another important factor in minimizing brain damage. When 20 × 10^5^ cells/kg were diluted in 0.5 mL or 1 mL and administered over 3 or 6 min, the denser cell concentration (0.5 mL) with slower injection (6 min) resulted in poorer functional outcomes [58]. Chua et al. also demonstrated that continuous arterial flow in the internal carotid artery during cell transplantation resulted in better recovery compared to when arterial flow ceased. Ge et al. found that smaller cell sizes yielded favorable results, with a 30 μm cell group causing arterial obstruction and acute ischemic damage, while 12 and 17 μm cell groups did not [65]. These findings highlight the importance of avoiding arterial obstruction during cell transplantation. As a result, the optimal dose for human treatment remains unknown; however, developing an optimal method to minimize arterial blood obstruction may be crucial for improving recovery outcomes.

#### 3.1.3. Transplantation Timing

Transplantation timing varies across studies, ranging from immediately after recanalization to day 14 (Figure 3). The most common transplantation timing occurs on day 1, followed by 1–6 h and 2–3 days. Immediate transplantation has become more common in recent years, likely due to the concept of treating patients directly after thrombectomy procedures. However, while thrombectomy has demonstrated that “the faster, the better,” this principle does not appear to apply to cell transplantation. Toyoshima et al. reported that stem cell transplantation at 24 h after recanalization resulted in better functional recovery compared to transplantation at 1 h and 6 h [53]. They also observed that the number of engrafted cells in the brain parenchyma was highest in the 24 h group, compared to the 1 h, 6 h, and 48 h groups. While the exact reason for this remains unclear, the authors speculated that the failure of ultra-early transplantation to show recovery may be due to the fact that the blood–brain barrier (BBB) remains functional up to 4 h after stroke, preventing transplanted cells from integrating into the damaged brain at this early stage [112,113]. Similar findings were reported by other researchers, who also found that transplantation at 24 h was superior to transplantation at 1 h after recanalization [60]. Rosenblum et al. also compared different time points (6 h, 1 day, 3 days, 7 days, and 14 days) of cell transplantation and found that cell engraftment was highest on day 3, followed by day 1, with other time points showing limited engraftment [74]. These results indicate that ultra-early cell administration may not be as beneficial as transplantation during the subacute phase, due to the detrimental environment. Later time points have also been evaluated in several studies. Mitkari et al. showed that both day 2 and day 7 administration resulted in better functional recovery compared to the control group, with the day 7 group showing enhanced angiogenesis at the infarction border [64]. Conversely, Ishizaka et al. reported that transplantation at earlier time points (1, 4, and 7 days) led to superior outcomes, with the 1-day group performing better than the 4-day and 7-day groups [71]. Based on these findings, ultra-early administration of stem cells may not be as beneficial as transplantation during the subacute phase, and optimal timing may be around 24–72 h after recanalization. However, further investigation into the precise mechanisms of action and the development of monitoring technologies is necessary to optimize the therapeutic potential for human clinical applications.

#### 3.1.4. Tracking Transplanted Cells and Visualizing Brain Condition

MR imaging, positron emission tomography (PET), single photon emission computed tomography (SPECT), laser Doppler/speckle imaging, and bioluminescence imaging are employed to track the fate of transplanted cells and visualize changes in the host brain (Appendix A). Through MR imaging, most studies evaluating the fate of transplanted cells have shown successful visualization, particularly in the ischemic core soon after cell transplantation. However, transplanted cells generally do not remain visible for longer than 72 h post-transplantation [32,36,52,82]. Other methodologies showed similar results, that bioluminescence imaging revealed that cells expressing luminescence rapidly decreased to approximately 40% on the next day of transplantation [50]. On the other hand, Andres et al. found that neural stem cells showed constant luminescence up to 48 h after transplantation [78]. The fate of transplanted cells in other organs is also examined by SPECT imaging; while the signal from the brain rapidly decreased 3–6 h after transplantation, the signal increased in the liver, spleen, and kidney [51,76,77]. However, the signal outside of the brain that truly represents live cells remains unknown because radioactive molecules released from dead and depredated cell particles containing radioactive molecules are likely to be captured in these organs. Regarding changes in the host brain, Huang et al. reported that cerebral blood flow (CBF), as evaluated by MR imaging, was upregulated in the cell treatment group at 28 days post-transplantation, with angiogenesis being enhanced [41]. The data was further confirmed by Du et al., who found increasing cerebral perfusion and glucose metabolism observed by SPECT and PET [62]. Bai et al. further reported visualization of the corticospinal tract after cell transplantation, showing that cell treatment successfully restored it, which was visualized through fractional anisotropy and diffusion tensor imaging [49]. Development of imaging technologies will enable a better understanding of the cells’ fate and the recovery mechanisms of the host brain.

#### 3.1.5. Mechanisms of Recovery

The intra-arterial transplantation of stem cells induces multiple recovery mechanisms, which can be classified into regeneration and damage reduction (Table 2). Regeneration encompasses neurogenesis, synaptogenesis, and angiogenesis, while damage reduction includes anti-inflammatory effects, anti-oxidative stress, anti-apoptotic effects, anti-ferroptotic effects, neuroprotection, endoplasmic reticulum protection, BBB protection, and exosome release.

Neurogenesis is one of the most important mechanisms for functional recovery. In addition to the direct differentiation of transplanted cells into neuronal cells [35,45,69], accelerated endogenous neurogenesis has also been observed. Transplanted cells enhance the maturation and migration of immature neural cells in the subventricular zone (SVZ) to the peri-damaged area [23,52,91]. Synaptogenesis is also observed, with endogenous neural cells demonstrating neurite and axon extension following stem cell transplantation [25]. Angiogenesis can occur either through the direct differentiation of transplanted cells into vessel-forming cells or by enhancing endogenous vessel formation, primarily through vascular endothelial growth factor (VEGF) released by the transplanted cells [23,26,41]. Angiogenic activity has been shown to restore not only histological vessel density but also cerebral blood circulation, as evidenced by brain perfusion studies around the peri-damaged area [24]. Interestingly, while angiogenesis is upregulated throughout the brain, distinct angiogenic gene expression profiles have been observed between the striatum, cortex, and hippocampus, indicating the complex angiogenic properties in different brain regions [25,34]. Cell modification with overexpression of very late antigen-4 (VLA-4) or neurogenin-1 increases cell adhesion to endothelial cells. This underscores the importance of the “stem cell-endothelial cell” adhesion in recovery [36,40].

Since most studies focus on acute-stage stem cell transplantation, damage reduction also plays a critical role in transplanted cell therapy. Stem cells mitigate local inflammation in the ischemic brain [27], as well as reduce oxidative stress [93], apoptosis [66], and ferroptosis [93] at the site of injury. Additional reports have demonstrated that specific brain components, such as neurons [27], the endoplasmic reticulum [30], and BBB [28] are protected by stem cell transplantation. Detailed functional analyses have revealed that inflammation is modulated by acid-sensing ion channels [29], apoptosis is alleviated by calcineurin released from transplanted cells [37], and BBB is preserved via protein kinase C delta (PKCδ)/aquaporin-4 (AQP4) pathway [28]. The upregulation of exosome release in the cell-treated group has been reported; however, their role in the brain remains incompletely understood and requires further investigation [23].

### 3.2. Traumatic Injury

Traumatic brain injury (TBI) affects over 50 million individuals worldwide each year, and a significant proportion of these patients experience long-term disabilities, as evidenced by the stagnation in return-to-work rates over the past 50 years [3]. Stem cell therapy is regarded as a promising approach to ameliorate functional deficits, and numerous basic and clinical trials have been conducted through intracerebral or intravenous transplantation [3]. Recently, following favorable results from our clinical trial, stem cell products employing intracerebral transplantation have been approved in Japan [114,115]. As such, the number of studies utilizing intra-arterial stem cell transplantation for TBI remains limited (Appendix A) [57,67,75,84]. The findings from these studies are similar to those of ischemic stroke research, where 5–25 × 10^5^ cells/kg of human or rat BMSC or NSC were injected 1–14 days post-TBI. While the data appear promising, further research is required to fully elucidate the therapeutic potential of stem cell therapy for TBI via intra-arterial transplantation.

### 3.3. Intracerebral Hemorrhage (ICH)

Due to the lack of effective therapies, stem cell transplantation can be a promising treatment modality for intracerebral hemorrhage (ICH). Similarly to traumatic brain injury, both intracerebral and intravenous stem cell transplantations have been the focus of research. Several clinical trials, including ours, are currently ongoing [1,2]. We identified only one study that utilized intra-arterial stem cell transplantation for ICH. Seyfried et al. reported that the injection of 10 × 10^5^ cells/kg of human BMSC one day after ICH, in combination with mannitol administration, successfully attenuated functional deficits by promoting neurogenesis [90].

### 3.4. Glioma

Malignant glioma is the most common and life-threatening primary brain malignancy in adults, with a median survival time of only 14 months, even with modern multidisciplinary treatments, including surgical resection followed by radiation and chemotherapy [116]. In this context, effective drug delivery to glioma cells is critical, and the ability of mesenchymal stem cells (MSCs) to accumulate in inflammatory regions presents a promising avenue for drug vector therapy. Oncolytic viruses, which have shown promising results in clinical trials [117], are being considered as part of this therapeutic approach. Therefore, MSCs infected with oncolytic viruses are emerging as a potential treatment strategy [117]. We identified four studies that focused on the treatment of glioma using virus/drug combinations for intra-arterial stem cell transplantation. In these studies, MSCs infected with oncolytic viruses successfully accumulated in the glioma tissue, but not in the healthy brain, when transplanted via the carotid artery. These cells inhibited tumor growth and increased survival time in animal models [80,86,118]. Conversely, MSCs overexpressing interferon gamma did not yield therapeutic benefits [33]. These findings suggest that transfected MSCs, which deliver infectious material, may have a higher potential for tumor treatment compared to the simple delivery of proteins.

### 3.5. Alzheimer’s Disease

Alzheimer’s disease (AD) is a progressive neurodegenerative disorder affecting over 55 million people worldwide, primarily in older adults. It causes memory loss, cognitive decline, and behavioral changes. Current treatments primarily focus on symptom management, utilizing cholinesterase inhibitors and emerging therapies targeting amyloid plaques, although no cure has been found. A single report investigating intra-arterial stem cell transplantation for AD has been published, though the results were unfavorable [92]. Since the BBB is not easily penetrable in AD, unlike in the acute and subacute phases of stroke or traumatic brain injury (TBI), the use of BBB-opening agents such as mannitol may prove beneficial and should be considered in future studies. Furthermore, due to the unlimited therapeutic window, multiple injections of stem cells or exosomes may be a good option for neurodegenerative diseases.

### 3.6. Parkinson’s Disease

Parkinson’s disease (PD) is a neurodegenerative disorder that affects approximately 10 million individuals worldwide. It is characterized by tremors, rigidity, bradykinesia, and postural instability. Although no cure exists, treatments such as levodopa and dopamine agonists help manage symptoms, and deep brain stimulation is used in advanced cases. Cerri et al. reported that mesenchymal stem cells (MSCs) pretreated with mannitol facilitated their passage into the damaged striatum and nigra. While functional recovery was observed with this approach, no modification of damaged brain cells was detected, suggesting that the therapeutic effect may be limited [55].

### 3.7. Safety Issues of IA Transplantation

Two reports have specifically focused on the safety issues related to intra-arterial transplantation, which is critical for enhancing its clinical application [58,65]. Cui et al. compared different cell doses and infusion speeds, concluding that slow injection of larger cell doses may lead to worse functional recovery due to the formation of microthrombi during the transplantation process. They suggested that dispersing cells more effectively might be necessary to reduce vascular obstruction [58]. Similarly, Ge et al. investigated the relationship between stem cell size and vascular obstruction, finding that smaller cell sizes (12–18 µm) caused less vessel obstruction and resulted in better functional recovery compared to larger cells (30 µm) [65]. They reported that 3D cell culture promoted smaller cell sizes compared to 2D culture, potentially reducing complications. Cell senescence has been shown to affect cell size, and supplementing culture media can influence this process. Our group recently discovered that human platelet lysate, compared to fetal bovine serum, resulted in better cell expansion rates and less senescence of smaller cell size, offering a potential improvement for stem cell culture conditions [119].

## 4. Preclinical Studies of Exosome Therapy

Recent advancements have highlighted that stem cell-derived exosomes may represent a pivotal therapeutic mechanism [7,8,9]. Exosomes are small vesicles (40–200 nm) released by stem cells and can be extracted from the supernatant of the culture medium. They encapsulate a variety of molecules, including DNA, mRNA, microRNA (miRNA), and proteins [6], all of which can be transferred into target cells, thereby providing therapeutic benefits in mitigating neurological diseases [7,8,9]. Furthermore, exosomes offer several advantages over traditional stem cell therapy, including high preservation capacity, low immunogenicity, the ability to cross the BBB, and the potential for drug encapsulation. As a result, exosomes are increasingly considered a promising alternative therapeutic approach [120]. Due to their novelty, the body of research on intra-arterial exosome infusion remains limited. (Appendix A) In the context of ischemic stroke, exosomes have been demonstrated to promote neurite outgrowth and exert anti-inflammatory effects. Xin et al. reported the efficacy of allogeneic BMSC-derived exosomes, in which BMSCs are overexpressed with microRNA (mir) 133b. They intra-arterially transplanted 3 × 10^11^ exosome particles suspended in 0.5 mL saline 24 h after middle cerebral artery occlusion. The exosomes enriched with miR-133b promoted neurovascular plasticity and achieved better functional recovery [94]. Similarly, traumatic brain injury was successfully attenuated by BMSC-derived exosomes [96]. Dabrowska et al. demonstrated that human BMSC-derived exosomes (1.3 × 10^9^ particles/1 mL) achieved better anti-inflammatory effect compared with BMSC itself. However, further evaluations of the therapeutic difference in exosomes and MSCs are not fully elucidated [96]. Lysosomal storage disorder is also evaluated with intra-arterial exosome transplantation. Seras-Franzoso et al. reported that mammalian cells overexpressing alpha-galactosidase A (GLA) successfully secrete exosomes containing abundant GLA. Exosomes were intra-arterially transplanted against GLA-deficient animals, and exosomes were rapidly uptaken, with approximately 7.5% of transplanted exosomes reaching the brain parenchyma and showing good enzyme activity in the brain [95]. Shi et al. further modified exosomes with a neuronal targeting peptide (rabies virus glycoprotein 29; RVG29), which facilitates cell delivery to the neuronal cells. Modified exosomes successfully enter the brain and attenuate apoptotic activity by affecting the p38/ERK signaling pathway [97]. Although further evaluation is necessary, intra-arterial exosome transplantation seems to be a promising therapeutic approach for brain disease.

## 5. Clinical Trials Using Stem Cell and Exosome via Intra-Arterial Transplantation

Twelve articles have investigated intra-arterial stem cell transplantation, while no clinical trials have been published on intra-arterial exosome transplantation [98,99,100,101,102,103,104,105,106,107,108,109]. The detailed data are presented in Table 3. Consistent with animal studies, the majority of these clinical trials focused on ischemic stroke, with additional studies addressing intractable diseases such as multiple system atrophy, epilepsy, and progressive supranuclear palsy. Unlike basic research, the clinical trials predominantly used autologous bone marrow mononuclear cells (BMMNCs), followed by autologous BMSC, likely due to safety concerns associated with early-phase clinical studies, which have prompted researchers to hesitate in using allogeneic cells. Nevertheless, to enable broader global distribution of stem cell products, the use of allogeneic stem cells remains a critical consideration. The cell doses administered in the trials varied, ranging from 3 to 160 × 10^5^ cells/kg, which is consistent with data from animal experiments as previously reported. Most studies report no serious adverse events associated with intra-arterial transplantation, although Giordano et al. observed asymptomatic ischemic signs in 85% (6 out of 7) of patients, which should not be overlooked [107]. Regarding ischemic stroke, while early single-arm clinical trials have shown favorable outcomes [99,102,104,109], large randomized trials have failed to demonstrate statistically significant differences in recovery [100,101,106,108]. In this context, Bhatia et al. reported a marginal trend toward recovery (*p* = 0.07), suggesting potential for future improvements. Notably, the transplantation cell dose in Bhatia’s study (83 × 10^5^ cells/kg) was higher than in other groups, indicating the possibility that a higher optimal cell dose for BMMNC intra-arterial transplantation may be necessary. Lee et al. and Chung et al., from the same research group, evaluated BMSC transplantation for multiple system atrophy and found that higher and medium-dose groups (6 or 9 × 10^5^ cells/kg) slowed disease progression compared to the low-dose group (3 × 10^5^ cells/kg), showing promise given the typically poor prognosis of this disease [98,103]. However, this research did not follow any basic studies concerning intra-arterial transplantation for this disease, and while the difficulty in mimicking animal models for rare diseases is acknowledged, a more detailed evaluation is essential before proceeding with clinical trials. Consequently, clinical trials employing intra-arterial stem cell transplantation remain in the early stages, and further, more refined trial designs are needed to validate their efficacy as demonstrated in animal models.

## 6. Conclusions and Future Direction

Intra-arterial stem cell/exosome transplantation offers several advantages over other transplantation approaches. Advances in catheter technology and technique will enhance the safety of this method, while a deeper understanding of the properties of stem cells and exosomes will facilitate improved functional outcomes for CNS diseases. Further evaluation of optimal cell sources, doses, timing, tracking methods, and underlying mechanisms is essential for the development of novel therapeutic modalities.

## Figures and Tables

**Figure 1 ijms-26-07405-f001:**
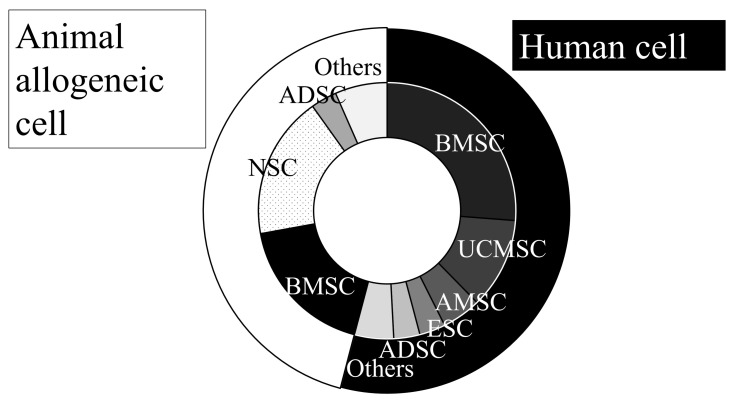
Cell sources used in animal experiment for intra-arterial transplantation.

**Figure 2 ijms-26-07405-f002:**
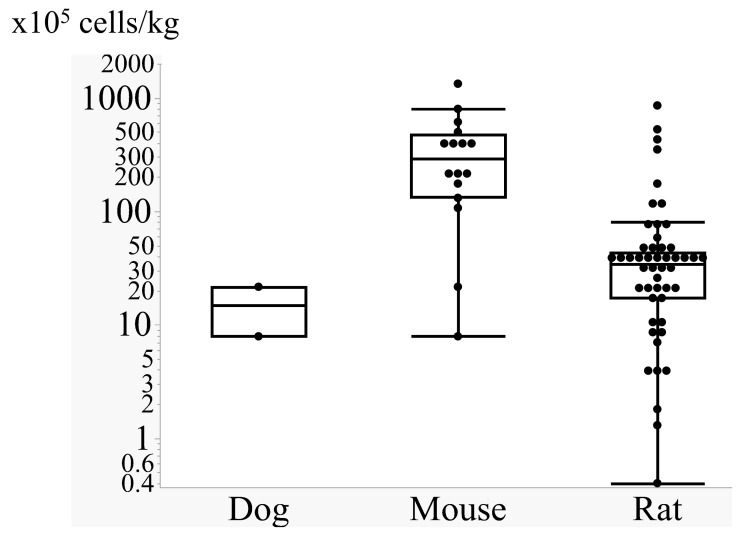
Cell doses used in animal experiments. Note that the Y-axis is on a logarithmic scale.

**Figure 3 ijms-26-07405-f003:**
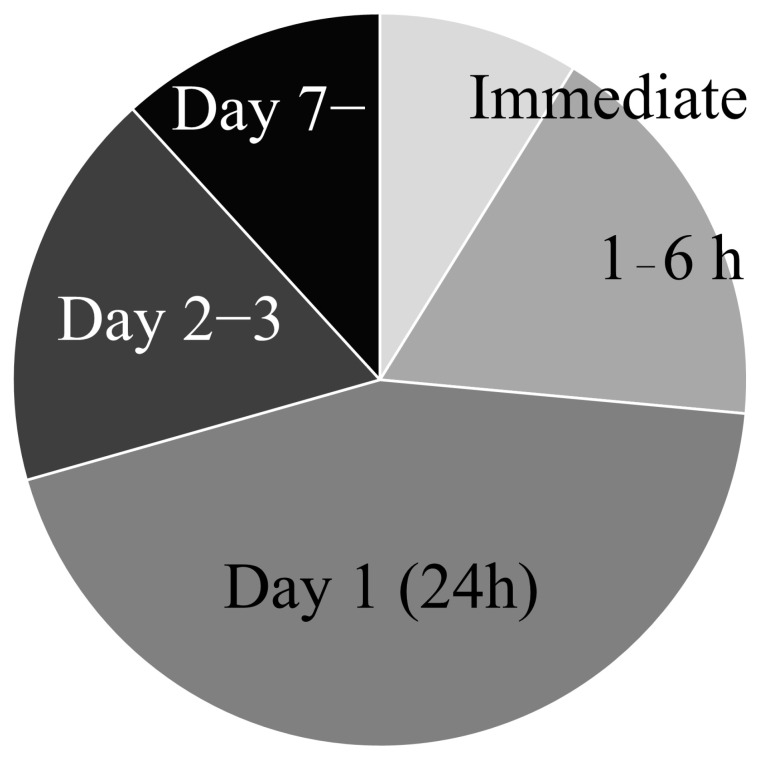
Transplantation timing of stem cell in the ischemic stroke model.

**Table 1 ijms-26-07405-t001:** Cell modification.

Methods	Modification Detail	Objective
Pretreatment	IL-1a	Anti-inflammatory effect
	neuregulin1	Neural differentiation
	MAPK inhibitor	Cell survival
	BDNF	Neural differentiation
Gene induction	Neural cell differentiation	Neural differentiation
	Integrin alpha 4	Cell adhesion
	The integrin Very Late Antigen-4 (VLA-4)	Cell adhesion
	Neurogenin	Neural differentiation
	CCL2	Cell migration to damaged area
Cell purification	CD34	Neurogenesis, angiogenesis
	CD133	Growth factor release

BDNF; brain-derived neurotrophic factor, CCL; C-C chemokine ligand, IL; interleukin, MAPK; mitogen-activated protein kinase.

**Table 2 ijms-26-07405-t002:** Mode of action.

Mode of Action	Detail of Action	Evaluated Factors
Regeneration	Neurogenesis	DCX, NeuN, MAP2, NGF, Nestin, SOX10, B-III tubulin, GFAP, Musashi1, Ki-67, Bur-U, Nogo-A, SYN, NF-200, NSE, Netrin-1, DCC
	Synaptogenesis	PSD95, Synaptophysin, GAP-43
	Angiogenesis	VEGF, HIF1a, Angiogenin, vWF, RECA, CD31
Damage reduction	Anti-inflammatory effect	IL-1b, IL-6, TNF-a, NLRP, IL-10, ED-1, MCP-1, Iba-1, CD45, IL-12, CD68, iNOS
	Anti-oxidative stress	GSH, MDA, Nitrite, catalase, Mitochondrial damage, ASIC1a, TBARS,
	Anti-apoptosis	caspase 3, caspase 12, TUNEL, FluoroJade C, bFGF, SDF-1a, Bcl-2
	Anti-ferroptosis	DMT1, TFR1, p53, SLC7A11, GPX4
	Neuro-protective function	SIRT-1, BDNF, NF-kb, neurotrophin-3, GDNF, HSP-27
	Endoplasmic reticulum-protective	GRP87, TrkB, p-eIF2α, ATF4, CHOP
	BBB-protective function	AQP4, PKC-d, MMP-9, VEGF
	Exosome release	CD63

ASIC1a; acid sensing ion channel 1a, AQP; aquaporin, BBB; blood–brain barrier, Bcl-2; B-cell/CLL lymphoma 2, BDNF; brain-derived neurotrophic factor, bFGF; beta-fibrous growth factor, CHOP; CCATenhancer-binding protein homologous protein, DCC; deleted in colorectal cancer, DCX; double cortin, DMT1; divalent metal transporter 1, GAP; growth associated protein, GDNF; glial cell-derived neurotrophic factor, GFAP; glial fibrillary acidic protein, GSH; reduced glutathione, HIF1; Hypoxia-Inducible Factor 1, HSP; heat shock protein, Iba-1; Ionized calcium-binding adapter molecule 1, IL; interleukin, iNOS; inducible nitric oxide, MAP2; microtubule-associated protein 2, MCP-1; Monocyte Chemoattractant Protein-1, MDA; malondialdehyde, MMP; matrix metalloprotease, NeuN; neuronal nuclear protein, NF-kb; nuclear factor-kappa B, NF-200; neurofilament-200, NGF; nerve growth factor, NLRP; Nucleotide-binding oligomerization domain, Nogo-A; neurite growth inhibition marker, NSE; neuron-specific enolase, PKC; protein kinase C, PSD95; postsynaptic density protein 95, SDF; soluble-derived factor, SIRT; Sirtuin, SYN; neurite growth marker, TBARS; thiobarbituric acid reactive substances, TFR1; transferrin receptor 1, TNF; tissue necrosing factor, TUNEL; terminal deoxynucleotidyl transferase dUTP nick end labeling, VEGF; vascular endothelial growth factor, vWF; von Willebrand factor.

**Table 3 ijms-26-07405-t003:** Stem cell IA clinical trial.

Author	PMID	Year	Phase	Disease	Number of Participants		Transplantation Route	Cell Source	Main Inclusion Criteria	Transplantation Timing	Endpoint Timing	IA Cell Doses, (×10^5^/kg *)		Cell Tracking	Main Conclusion
					Patients	Control						Total Cell Numbers	Cell Dose (×10^5^/kg)		
Battistella et al. [102]	21175286	2011	I	ischemic stroke	6	-	IA	Autologous BMMNC	NIHSS; 4–17	2–3 months	6 months	1–5 × 10^8^	16–83 × 10^5^/kg	99m-Tc	Feasible and safe, cells soon distributed in the liver
Friedrich et al. [99]	22507676	2012	I/II	ischemic stroke	20	-	IA	Autologous BMMNC	NIHSS > 8	3–7 days	6 months	5–60 × 10^7^	8–100 × 10^5^/kg	-	Feasible and safe, satisfactory clinical improvement occurred in 30% of patients
Moniche et al. [101]	22764211	2012	I/II	ischemic stroke	10	10	IA	Autologous BMMNC	NIHSS 15.6 (mean)	5–9 days	6 months	1.6 × 10^8^ (mean)	26 × 10^5^/kg	-	Safe, but no difference regarding the functional recovery was seen compared with control group
Lee et al. [103]	22829267	2012	II	multiple-system atrophy	11	16	IA and IV	Autologous BMSC	UMSRS 30–50	-	12 months	4 × 10^7^	7 × 10^5^/kg	-	Functional recovery and MRI findings were significantly better in treatment group
Banerjee et al. [104]	25107583	2014	I	ischemic stroke	5	-	IA	Autologous BMMNC (CD34+)	NIHSS > 8	7 days	6 months	1.2–2.7 × 10^6^	2–5 × 10^5^/kg	-	Feasible and safe
DaCosta et al. [105]	27688159	2018	I/II	temporal lobe epilepsy	20	-	IA	Autologous BMMNC	Medically refractory epilepsy	-	6 months	1–10 × 10^8^	16–160 × 10^5^/kg	-	Feasible and safe, 40% of the patients became seizure-free after transplantation
Bhatia et al. [106]	29545253	2018	II	ischemic stroke	10	10	IA	Autologous BMMNC	NIHSS > 7	8–15 days	6 months	5 × 10^8^	83 × 10^5^/kg	-	Feasible and safe, better trend of recovery for treatment group (*p* = 0.07)
Savitz et al. [100]	30586746	2019	II	ischemic stroke	29	17	IA	Autologous BMMNC (ALDH+)	mRS > 3	9–15 days	3 months	1.6–7.5 × 10^7^	3–13 × 10^5^/kg	-	No statistical differences were seen between treatment and control groups
Hammadi et al. [109]	30777565	2019	I	ischemic stroke	37	0	IA	Autologous BMMNC	MCA territory	3 months–5 years	6 months	5.0–6.0 × 10^8^	83–100 × 10^5^/kg	-	67% of patients showed functional recovery
Chung et al. [98]	34712335	2021	I	multiple-system atrophy	9	0	IA	Autologous BMSC	UMSRS 30–50, Disease duration < 5 years	-	3 months	-	3, 6, 9 × 10^5^/kg	-	Feasible and safe, medium and high dose groups showed a slower increase in UMSARS score than low group
Giordano et al. [107]	34712113	2021	I	progressive supranuclear palsy	7	0	IA	Autologous BMSC	PSP diagnosis criteria	-	12 months	77–156 × 10^6^	10–20 × 10^5^/kg	-	Asymptomatic abnormal signs were found in the MRI, no significant functional recovery
Moniche et al. [108]	36681446	2023	II	ischemic stroke	37	36	IA	Autologous BMMNC	NIHSS 6–20	1–7 days	6 months	-	0, 20, 50 × 10^5^/kg	-	No statistical differences were seen between treatment and control groups

ALDH; aldehyde dehydrogenase, BMMNC; bone marrow mononuclear cell, BMSC; bone marrow-derived stem cell, IA; intra-arterial, IV; intravenous, NIHSS; National Institute of Health stroke scale, UMSARS; Unified multiple system atrophy rating scale. * body weight not indicated are calculated as 60 kg.

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
