# Peer review of "Intra-Arterial Administration of Stem Cells and Exosomes for Central Nervous System Disease"

_ijms, 2025, doi:10.3390/ijms26157405_

Round 1
Reviewer 1 Report
Comments and Suggestions for Authors
The current manuscript presents a very thorough review of the therapy based on intra-arterial delivery of stem cells and their exosomes against brain injury, including stroke, TBI, AD, and PD. The literature reviewed is up to date, and the flow of the presentation is smooth. The summary of the current status of SC/exosome treatment, their efficacy, and remaining challenges is valid. Overall, this review is good for the readers, although it would benefit from the addition of some ideas or visions for future preclinical and clinical tests. There are a few minor questions that should be clarified before its acceptance.
- Comparison to i.v. delivery: “Intra-56 venous transplantation is minimally invasive but results in limited delivery of cells and 57 exosomes to the brain, which may compromise the therapeutic efficacy.15 Intranasal ap-58 proach is also minimally invasive, however, their absorption rate is not as high as ex-59 pected.13”
There are a few papers that have examined how much of i.v. delivered SCs can be detected in the brain. It will be better to include the quantitative information for a better understanding.
- Improve the penetration of SCs and exosomes across the BBB: Although the BBB is transiently but readily opened in the acute and subacute phases of stroke and TBI, in the cases of AD and PD, BBB opening reagents, like mannitol, should be used before the SC/exosome delivery. This point can be discussed.
- Most of the SC/exosome delivery is performed in the acute/subacute phase post-injury. However, whether the subject can benefit from repetitive delivery in their chronic phase, or those with chronic CNS conditions (AD/PD) can also benefit from multiple injections, remains largely unknown and should be discussed/suggested.
There are a few minor grammar errors, like this rollign sentence: “Exosomes, ranging from 40-200 nm, can be ex-351 tracted from the culture medium supernatant, which these vesicles encapsulate a variety 352 of molecules, including DNA, mRNA, microRNA (miRNA), and proteins,6 all of which 353 can be transferred into target cells, thereby conferring therapeutic benefits in mitigating 354 neurological diseases.7-9”
Reviewer 2 Report
Comments and Suggestions for Authors
The authors summarise the basic and clinical research of the intra-arterial administration of stem cells and exosomes in brain pathologies. The review article is well written, easy to read, and contains an adequate amount of previously published data. The authors have listed accurately and in detail the differences in the study designs they are reviewing. The number of citations, 112, is adequately high for the review article.
I would like to ask the authors to explain why they have prepared the review article. How does the scientific community benefit from this manuscript?
Secondly, the reviewers should present some criticism. E.g., in Chapter 3.4 Glioma, the authors are reporting the studies according to which:” MSCs infected with oncolytic viruses successfully accumulated in the glioma tissue, but not in the healthy brain”.
Author Response
please sse the attachment
